# Impact of the COVID-19 Pandemic on Incidence and Observed Survival of Malignant Brain Tumors in Belgium

**DOI:** 10.3390/cancers16010063

**Published:** 2023-12-21

**Authors:** Tim Tambuyzer, Dimitri Vanhauwaert, Tom Boterberg, Steven De Vleeschouwer, Hanna M. Peacock, Joanna Bouchat, Geert Silversmit, Freija Verdoodt, Cindy De Gendt, Liesbet Van Eycken

**Affiliations:** 1Belgian Cancer Registry, 1210 Brussels, Belgium; tim.tambuyzer@kankerregister.org (T.T.); joanna.bouchat@registreducancer.org (J.B.); elizabeth.vaneycken@kankerregister.org (L.V.E.); 2Department of Neurosurgery, AZ Delta Hospital, 8800 Roeselare, Belgium; 3Department of Radiation Oncology, Ghent University Hospital, 9000 Ghent, Belgium; tom.boterberg@ugent.be; 4Department of Neurosurgery, UZ Leuven, 3000 Leuven, Belgium; steven.devleeschouwer@uzleuven.be; 5Laboratory of Experimental Neurosurgery and Neuroanatomy, Department Neurosciences, LEUVEN BRAIN INSTITUTE (LBI), KU Leuven, 3000 Leuven, Belgium

**Keywords:** COVID-19, brain tumors, glioma, incidence, survival, pandemic

## Abstract

**Simple Summary:**

A decline in cancer diagnoses during the COVID-19 pandemic has been reported in many countries, including Belgium. Most of these reports focus on cancers with the highest incidence, while data for brain tumors are scarce. This study evaluates the impact of the COVID-19 pandemic on the incidence, treatment strategies, and observed survival of adults diagnosed with malignant brain tumors in Belgium in 2020. The results of this study, confirming a dramatic impact on incidence and survival, should be taken into account by policy makers when implementing measures during future disease outbreaks. Pathways for (neuro)-oncological care should be continued, and if necessary adapted, and physicians should pro-actively develop frameworks for shared and broadly informed decision making when capacity for care is reduced.

**Abstract:**

(1) Background: This study evaluates the impact of the COVID-19 pandemic on the incidence, treatment, and survival of adults diagnosed with malignant brain tumors in Belgium in 2020. (2) Methods: We examined patients aged 20 and older with malignant brain tumors (2004–2020) from the Belgian Cancer Registry database, assessing incidence, WHO performance status, vital status, and treatment data. We compared 2020 incidence rates with projected rates and age-standardized rates to 2015–2019. The Kaplan–Meier method was used to assess observed survival (OS). (3) Results: In 2020, there was an 8% drop in age-specific incidence rates, particularly for those over 50. Incidence rates plunged by 37% in April 2020 during the first COVID-19 peak but partially recovered by July. For all malignant brain tumors together, the two-year OS decreased by four percentage points (p.p.) in 2020 and three p.p. in 2019, compared to that in 2015–2018. Fewer patients (−9 p.p.) with glioblastoma underwent surgery, and the proportion of patients not receiving surgery, radiotherapy, or systemic therapy increased by six percentage points in 2020. (4) Conclusions: The COVID-19 pandemic profoundly impacted the diagnosis, treatment strategies, and survival of brain tumor patients in Belgium during 2020. These findings should guide policymakers in future outbreak responses, emphasizing the need to maintain or adapt (neuro)-oncological care pathways and promote informed decision making when care capacity is limited.

## 1. Introduction

A decline in cancer diagnoses during the COVID-19 pandemic has been reported in many countries, including Belgium [1,2,3,4,5]. Since healthcare facilities had to deviate much of their efforts and resources towards the care of COVID-19 patients, access to conventional healthcare, including oncological care, was significantly impaired. Therefore, healthcare services that were deemed non-essential, including cancer screening, were suspended in Belgium—and many other countries—for several weeks [6,7,8,9]. Most of these reports focus on cancers with the highest incidence, while data for brain tumors are scarce.

The most common brain tumor subtypes in Belgium are glioblastoma and hematolymphoid tumors involving the CNS, respectively representing 67% and 7% of all malignant brain tumors in adults in Belgium in 2015–2019, with a corresponding yearly average of 573 and 64 new diagnoses, respectively. Hematolymphoid tumors involving the CNS are mainly diffuse large B-cell lymphomas (DLBCL; 87%). Since glioblastoma and DLBCL of the brain have unfavorable prognoses, in combination with a steeply decreasing survival curve in the first few months after diagnosis, it is expected that short term delays in diagnosis and subsequent treatment, or divergence from recommended treatment strategies, might affect outcomes for these cancer types [10,11]. Here, we provide a nationwide analysis that aims to quantify and evaluate the impact of the COVID-19 pandemic on the incidence of malignant brain tumors in adults in Belgium, as well as on treatment strategies and observed survival (OS).

## 2. Materials and Methods

Since 2004, the Belgian Cancer Registry (BCR) has been legally responsible for the collection of data on all new oncological diagnoses. Its database is estimated to be more than 95% complete from 2004 onwards, thanks to mandatory cancer registration in Belgium [12,13]. The inclusion criteria for the selection of patients from the BCR database for this study were as follows: an age of 20 years or older at the time of diagnosis, Belgian residence at the time of diagnosis, International Classification of Diseases for Oncology, 3rd Edition (ICD-O-3) topography code C71 (brain), ICD-O-3 behavior code/3 (malignant, primary), and relevant ICD-O-3 morphology codes (see Appendix A) [14]. The vital status of patients was updated until 31st December 2022. 

Cancer incidence was described as the number of new diagnoses, age-specific (in 5-year-wide groups), and age-standardized incidence rates (WSR: World Standardized Incidence Rate). The observed survival probability was estimated using the Kaplan–Meier method [15]. The WHO performance status and treatment data were extracted from multidisciplinary oncological consult (MOC) forms. These forms contain information on planned and/or given treatment categories. In the context of this study, proportions of patients were calculated for each of the following treatment categories: surgery, radiotherapy, systemic therapy (which, in this study, refers to chemotherapy, targeted therapy, and immunotherapy), and patients who received none of these.

At the year level, the impact of the COVID-19 pandemic on cancer incidence was evaluated by comparing the observed incidence rates of 2020 with projected incidence rates for 2020. The latter were based on an extrapolation of the incidence trends over the period 2004–2019 [16]. At the month level, age-standardized incidence rates for 2020 were compared to the average incidence at the month level for 2015–2019. In all month-level comparisons, the months January and February 2020 were considered pre-COVID-19 baseline months. Besides results for all malignant brain tumors combined, more detailed analyses were added for the two most frequent subtypes of malignant brain tumors: glioblastoma and hematological malignancies. Two-sided z-tests, at a 5% significance level, were used for all statistical comparisons of incidence and survival between different months/years. Since the observed survival for patients diagnosed in 2019 might have been impacted by the COVID-19 pandemic in 2020, the OS in 2020 was compared to the OS in 2019 and 2015–2018, separately.

## 3. Results

### 3.1. Incidence of Malignant Brain Tumors

The observed age-specific incidence rates for malignant brain tumors in 2020 were similar compared to those in 2015–2019 for patients under 50 years of age, but were 8% lower for the age group 50+ (Figure 1). As such, this study further focused on the 50+ age group. The age-standardized incidence rate (WSR; Figure 2) in the 50+ age group for 2020 (13.0/100,000) was significantly lower than the projected incidence for 2020 (14.9/100,000) (*p* < 0.005), corresponding with an estimation of 70 missed diagnoses (*N_Observed_* = 640; *N_Projecte_*_d_ = 710). Moreover, the observed incidence rate for malignant brain tumors in Belgium in 2020 was the lowest ever noted since the start of the registrations in 2004. After pre-COVID-19 baseline months (January–February 2020), the standardized incidence rates showed a significant decrease of −37% (*p* < 0.005) in April 2020 vs. April 2015–2019 (Figure 3A), which coincided with the peak of the first COVID-19 wave in Belgium. A subsequent recovery phase of delayed diagnoses can be noted, with a modest rebound in July 2020 (although the difference with July 2015–2019 was not significant; *p* < 0.10). In addition, the proportion of patients presenting with a WHO performance score of 2–4 (i.e., symptomatic and partially or fully bedbound) at diagnosis was 10 p.p. higher in 2020 compared to that in previous years (Figure 3B), and in April and November 2020—the first and second COVID-19 peaks—approximately 50% of all patients presented with a WHO performance score of 2–4.

### 3.2. Survival of Patients with Malignant Brain Tumors and Subtypes

OS for patients diagnosed with a malignant brain tumor by month in 2020 varied depending on the phase of the pandemic (Figure 4A). For patients diagnosed prior to the pandemic, in January–February 2020, and thus likely to have received at least part of their oncologic care during the first COVID-19 wave, the OS curve initially coincides with the curve of 2015–2018. However, at two years after diagnosis, the observed survival is reduced by 6 p.p. (17% for patients diagnosed in January–February 2020 vs. 23% for 2015–2018). For patients diagnosed with brain tumors in March–April 2020, during the first peak of the pandemic, the OS curve started to diverge from the 2015–2018 curve 5 months after diagnosis, to end with a difference of −10 p.p. at 2 years after diagnosis (13% for Apr–March 2020). During those two months of the first peak of the pandemic, the proportion of patients diagnosed with glioblastoma (78%) was higher compared to that in 2015–2018 (74%) and the other months of 2020 (Figure 4B). Patients diagnosed in July 2020, corresponding to the diagnostic recovery peak in the monthly incidence trend curve, had the lowest OS results in the first four months after diagnosis, but had more similar 2-year OS (19%) as compared to that in 2015–2018. Figure 5 presents OS at 0.5, 1, 1.5 and 2 years for patients diagnosed in 2020, 2019 and 2015–2018, by subgroups. For hematolymphoid malignancies (approximately 93% DLBCL in 2020), a strong impact of the COVID-19 pandemic on the OS in 2020 was marked relative to that in 2015–2018, with significant reductions of −16 p.p. by 6 months (47% vs. 63%), −18 p.p. at 1 year (36% vs. 54%), and −17 p.p. at 2 years after diagnosis (29% vs. 46%). For glioblastoma, there was a smaller but significantly lower 2-year OS (−3 p.p.) for patients diagnosed in 2019 when compared with that in 2015–2018. Also, for patients diagnosed in 2020, a decreasing trend was noted (−2 p.p.), although it was not significant. For all malignant brain tumors together, the 2-year OS significantly decreased in 2020 (−4 p.p.), as well as in 2019 (−3 p.p.), compared to that in 2015–2018.

### 3.3. Treatment for Glioblastoma and Hematolymphoid Tumors 

Based on information on given and/or planned treatment from MOC registration files, for glioblastoma diagnosed in 2020, a decrease in the number of surgeries was noted when compared to that in 2015–2018 (−9 p.p.). For (radio)chemotherapy in glioblastoma (including radiotherapy with concomitant Temozolomide), no notable change was observed. In hematolymphoid malignancies (mainly DLBCL), the proportion of patients receiving systemic therapy increased by +8 p.p. for patients diagnosed in 2019 vs. 2015–2018, but decreased again by −10 p.p. in 2020. For all types of malignant brain tumors combined, the proportion of patients who received none of these three treatment categories (i.e., surgery, radiotherapy and systemic therapy) increased for patients diagnosed in 2020 (+6 p.p.) and 2019 (+3 p.p.) as compared to that in 2015–2018.

## 4. Discussion

The impact of the COVID-19 pandemic on both observed incidence and survival from malignant brain tumors in patients over 50 years in Belgium is clearly demonstrated in this study. The decreased incidence combined with the less favorable WHO performance (WHO 2–4) status at diagnosis in March–April 2020 vs. that in 2015–2019 suggests a delay in diagnosis, especially for patients with moderate to no symptoms. Unlike for some other malignancies, a delay in the diagnosis and treatment of brain tumors can be associated with a rapid deterioration of neurological function [17]. Castaño-Leon reported that one-quarter of patients experienced clinical or radiological deterioration while on a waiting list for surgery based on a multicenter study in Spain (size of study population: 680 patients) [18]. Single-center studies in Austria and China reported a larger-than-average tumor volume upon surgery during the COVID-19 pandemic, also suggesting a potential delay in diagnosis [19,20,21].

The worse WHO performance scores during March-April 2020 suggest that relatively more patients with severe symptoms were diagnosed during the first COVID-19 wave in Belgium. However, not only in March–April 2020, but also in most months during the pandemic, the proportion of patients with a weakened condition at diagnosis (a WHO performance score of 2–4) was higher compared to that in 2015–2019, suggesting that there has consistently been a small proportion of patients with delayed diagnosis since the start of the pandemic (Figure 3). From mid-March, all (deemed) non-essential healthcare services were temporarily halted in Belgium due to a governmental decision, limiting the accessibility of medical consultations and imaging [1]. One Belgian academic pathology department reported a reduction (although this was not statistically significant) in central nervous system (CNS) tissue samples received in March and April 2020 [1,22]. Moreover, the healthcare crisis also caused the reluctance of patients to seek medical care [4,23,24,25,26]. In addition, for patients already diagnosed, a delay in treatment and follow up amplifies the uncertainty that generally already comes with a cancer diagnosis in a normal pre-COVID-19 context [25]. As the first COVID-19 peak ended, access to healthcare was gradually restored—as in other countries—and in July 2020, a recovery phase took place [8].

The 2-year OS for malignant brain tumors diagnosed in 2020 is significantly lower compared to that in previous years (Figure 4 and Figure 5). One factor could be that a proportion of the diagnoses were delayed [22,27]. The steep initial decrease in the OS curve in July 2020 suggests that there may be a relation between delayed diagnoses and impacted survival. Another likely factor is a change in treatment strategies, as reported by others, and as some of our results also indicate [18,28]. The observed decrease in the 2-year OS, not only for patients diagnosed in March–April 2020 (i.e., a higher proportion of patients with glioblastoma and more patients with severe symptoms as quantified via the WHO performance score), but also for those diagnosed in January-February 2020, supports this hypothesis. Moreover, for patients diagnosed with glioblastoma in 2019 (the year before the beginning of the pandemic and thus patients who were partially treated or followed up during the pandemic), we also note a significantly decreased OS of −3 p.p. compared to that in 2015–2018. In 2020, the proportion of patients treated with surgery for glioblastoma was 9 p.p. lower compared to that in 2015–2018. This can probably be explained by the reduced surgical and subsequent post-surgical intensive care capacity at the pinnacle of the pandemic. Furthermore, the fact that some patients were diagnosed with a delay, i.e., in a worse condition, may have shifted decisions towards more supportive care. A survey among 15 neuro-oncology centers in the U.K. illustrates that 9% of patients with high-grade glioma, who would have been offered surgery as standard pre-COVID-19 care by the multidisciplinary board, were given a different recommendation such as radiotherapy without biopsy (4%), or supportive care (5%), because of resource and capacity restrictions. Moreover, for newly diagnosed high-grade glioma patients to be treated with chemotherapy, 30% were offered best supportive care or a delay in treatment instead [29]. It is, however, not clear whether these deviations in standard care impacted survival rates in our cohort. Norman et al. reported significantly more delays in the care and use of telehealth in a single-center cohort of neuro-oncology patients in 2020 as compared to those in 2019, but noted no differences in outcome between 2020 and 2019 [30]. Ideally, those outcomes should have been compared with the outcomes in earlier years, since for some patients diagnosed in 2019, treatment plans and clinical decision making might have been impacted by the COVID-19 pandemic in spring 2020. In this context, Bernhardt et al. proposed a set of recommendations mainly focusing on adjuvant treatments for patients with high-grade gliomas to be treated with during the healthcare crisis, such as hypofractionated radiotherapy or withholding Temozolomide in unmethylated MGMT patients [31]. It is, however, not clear to what extent these recommendations were followed in Belgium during different phases of the pandemic. However, a final factor that might have influenced survival in brain tumor patients is an intercurrent COVID-19 infection causing a delay in or suspension of treatment, or even death [21,32].

In an earlier study of the Belgian Cancer Registry, we concluded that a substantial excess mortality was observed in the prevalent Belgian cancer cohort (diagnosed between 2013 and 2018 and alive on 1 January 2020) during the first wave of COVID-19, which, however, was not considerably different from the excess mortality in the general population [33]. Based on administrative information from reimbursement data, we found that at least 9.9% of our study population for 2020 suffered from a concomitant COVID-19 infection that was confirmed in a hospital setting during the follow up of our study. Such COVID-19 infections can have an impact on the treatment trajectories and observed survival of these patients, but we cannot draw strong causal conclusions since cause of death information obtained from death certificates was unavailable at the time of this study. Additionally, it could also be possible that some brain tumor diagnoses were not even registered due to the excess mortality caused by the COVID-19 pandemic in the general population. Based on a statistical approximation using the number of deaths of the general Belgian population in 2020, this impact was estimated to be negligible in our study population (<1 missing diagnoses) [34].

The OS of patients diagnosed with hematolymphoid malignancies involving the CNS (mainly DLBCL) in 2020 shows a substantial decrease already in the first few months after diagnosis (−16 p.p. at 6 months), and this decline grew larger in the months thereafter (to −18 p.p. at 1 year and −17 p.p. at 2 years). In addition, in 2020, the proportion of these patients receiving systemic therapy was 10% lower as compared with that in 2019. The larger decline in the OS of hematolymphoid malignancies involving the CNS (predominantly DLBCL) compared with glioblastoma could be explained by a typically higher number of hospital stays during the intensive treatment trajectory for patients diagnosed with DLBCL (treated with intravenous chemotherapy and sometimes requiring protective isolation, instead of oral Temozolomide for patients with glioblastoma). Treatments for DLBCL are thus more at risk to be affected by the immense pressure on the hospitals due to the COVID-19 outbreak. On the other hand, a subtle increase in 6-, 12-, and 18-month OS is noted for patients with DLBCL diagnosed in 2019 as compared to that in earlier years (Figure 5). This can probably be explained by scientific progress and the subsequent augmented availability of new (reimbursement) pharmaceuticals on the market [35].

A limitation of this study is the relatively small number of patients in some of the analyses, particularly for estimates of subtypes of brain malignancies and at the month level (e.g., the number of patients at risk of hematological malignancies was 58 in 2020). Secondly, information on treatment modalities is based on data extracted from the multidisciplinary oncological consult (MOC) forms. These forms allow the rapid evaluation of treatment modalities with a delay of about 6–18 months after diagnosis, yet they only provide information on treatment that is planned and/or given at the time of registration in the cancer registry, and analyses based on the data should therefore be considered as hypothesis-generating. Our results based on MOC data were, however, in line with preliminary results based on administrative information from reimbursement data. These comparisons confirmed the observed trends. However, these reimbursement data of therapeutic procedures were not fully complete at the time of analyses and no information on clinical trial treatment is available in these data. In many hospitals, inclusion in clinical trials was suspended during the pandemic, but on the other hand, very few clinical trials were open for brain tumor patients in that period. Another limitation of this study is that there are no other measures for the severity of disease (compared to other malignancies) than the diagnosed subtype and the WHO performance score. Based on our nationwide results and the limited data on neuro-oncological care access and strategies in the literature, it might be expected, however, that similar findings on survival will be observed by other cancer registries worldwide.

## 5. Conclusions

A clear impact on the incidence and survival of malignant brain tumors during the COVID-19 pandemic has been observed, including a significant drop in incidence rates in April 2020 (−37%) compared to those in 2015–2019, and a significant decrease of −4 p.p. in the 2-year OS in 2020, as well as in 2019 (−3 p.p.), compared to that in 2015–2018. This should be taken into account by authorities when implementing measures during future infectious outbreaks or other crises with an impact on health care resources. Pathways for (neuro)-oncological care should be continued, and if necessary adapted, but complete suspension is not justifiable. Moreover, physicians should pro-actively develop skills, flows, and frameworks for shared and broadly informed decision making when the capacity for conventional care is impacted.

## Figures and Tables

**Figure 1 cancers-16-00063-f001:**
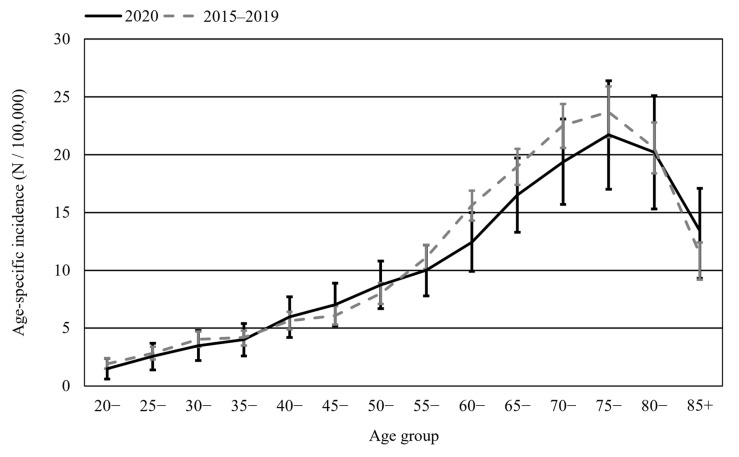
Age-specific incidence rates (N/100,000; 5-year age groups) with 95% confidence intervals for malignant brain tumors in adults (patients aged 20 years and older) for incidence years 2020 and 2015–2019.

**Figure 2 cancers-16-00063-f002:**
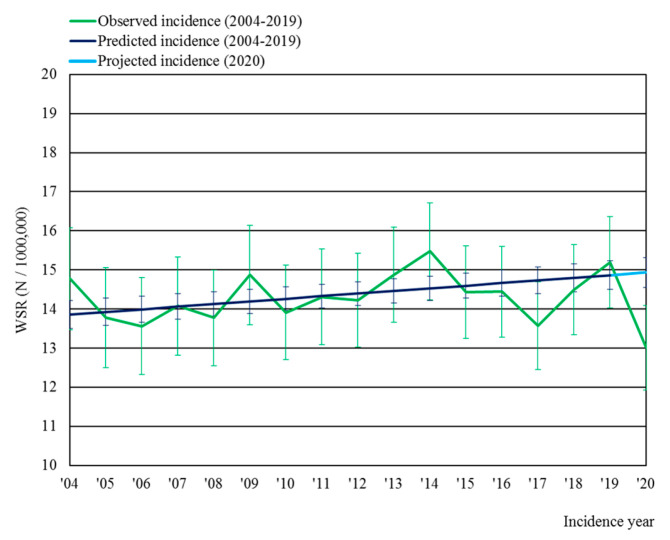
Observed age-standardized incidence rates (WSR, with corresponding 95% confidence intervals) versus predicted and projected incidence (WSR, with corresponding 95% confidence intervals) for malignant brain tumors in the age group 50+ in Belgium.

**Figure 3 cancers-16-00063-f003:**
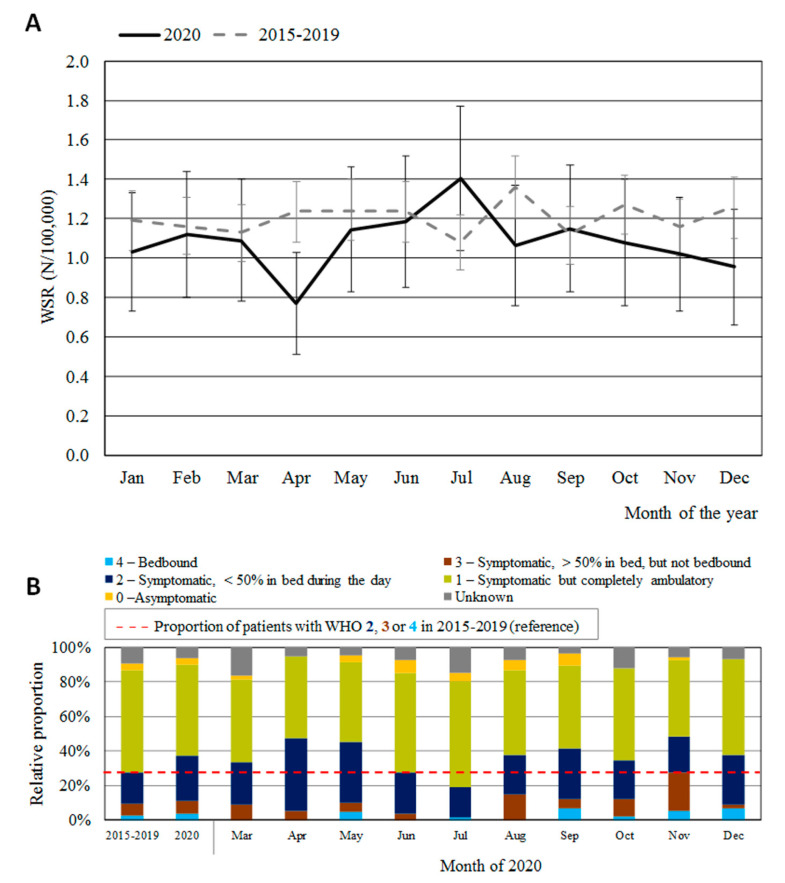
All malignant brain tumors, age group 50+. (**A**) Age-standardized incidence rates (WSR) and 95% confidence intervals by month for 2020, compared to the average month level for 2015–2019. January and February can be considered baseline months. (**B**) WHO performance status at diagnosis. Horizontal red dashed line indicates the proportion of patients presenting with a WHO status of 2–4 in the reference period 2015–2019.

**Figure 4 cancers-16-00063-f004:**
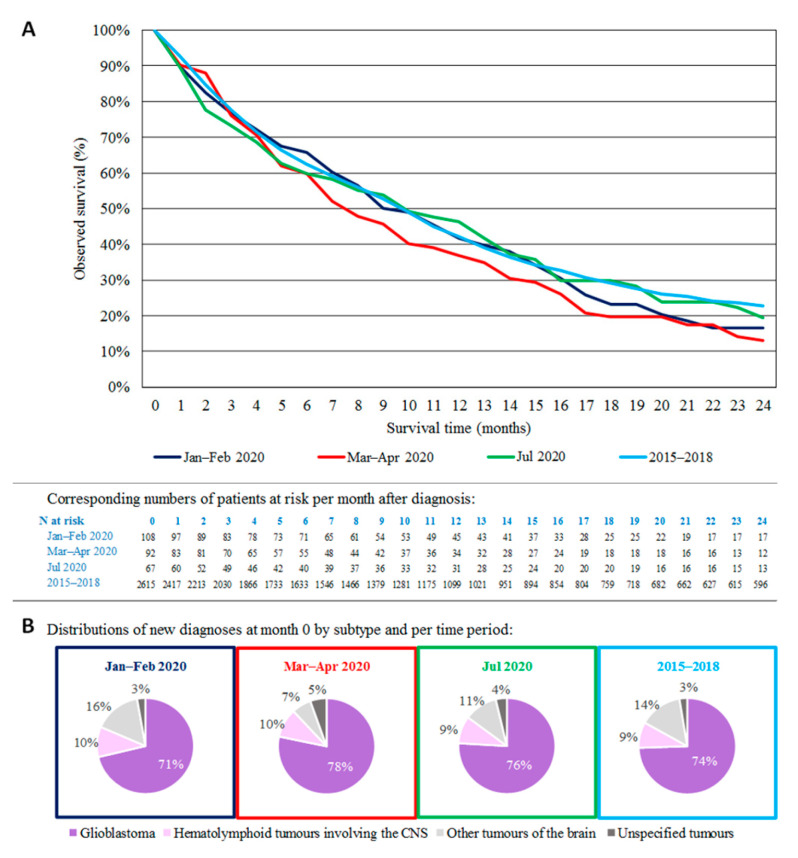
(**A**) Observed survival curves for malignant brain tumors in the age group 50+ during the year 2020 vs. that in 2015–2018 in Belgium and the corresponding numbers of patients (N) at risk. In this graph, OS results are calculated on the month level with the Kaplan–Meier method and the points obtained are connected with a line. (**B**) The corresponding distribution of tumor types during the year 2020 vs. that in 2015–2018.

**Figure 5 cancers-16-00063-f005:**
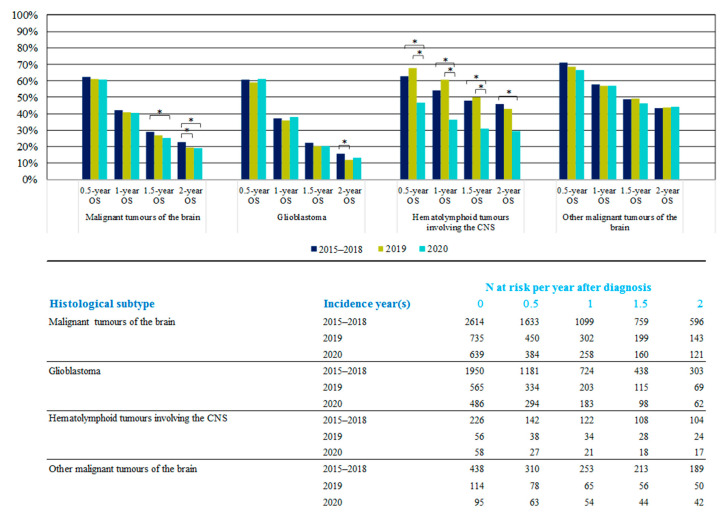
Diagram showing 0.5-, 1-, 1.5- and 2-year OS (observed survival) for subgroups of malignant brain tumors during 2020 vs. 2019 and 2015–2018. Significant differences (*p* < 0.05) are indicated with an asterisk (*).

## Data Availability

The cancer cohort data used and analyzed during the study are available upon reasonable request and those seeking them should be directed to info@kankerregister.org. The pseudonymized data can be provided within the secured environment of the Belgian Cancer Registry after having been guaranteed that the applicable GDPR regulations are applied.

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
