# Peer review of "Impact of the COVID-19 Pandemic on Incidence and Observed Survival of Malignant Brain Tumors in Belgium"

_cancers, 2023, doi:10.3390/cancers16010063_

Round 1

Reviewer 1 Report

Comments and Suggestions for Authors

The topic discussed by the authors is relevant and well suited for this journal.

  1. The article addresses the management of malignant tumors during the COVOD-19 period. 
  2. The topic is relevant and important for clinical practice. The article is well written and the topic well justified. 
  3. The article brings insight of specific patient management during the COVID-19 times. 
  4. The article has no flaws, according tommy opinion. It is nicely written and cam be accepted in its present form. 
  5. The main topic is well presented, as are the Discussion and the Conclusions. The main question is well addressed. 
  6. The references are appropriate. 
  7. The pictorial material is adequate, figures and tables are ok. 

The manuscript brings new information regarding malignant tumours and their management during COVD-19. The paper is well written and no canges are necessary. I have no suggestions to add. I recommen accepting the manucript as written. 

Author Response

[ANSWER]:  Thank you very much for your positive and constructive feedback.

Reviewer 2 Report

Comments and Suggestions for Authors

The authors report incidence and survival of malignant brain tumors in Belgium using national database. It's meaningful to analyze the impact of COVID 19 pandemic. Their discussion on treatment delay and a change in treatment strategy due to hospital capacity as a cause of lower observed survival is reasonable. It would be interesting to add data on metastatic brain tumors for comparison, but I suppose the article is well-written and should be accepted with minor language editting.

Comments on the Quality of English Language

Almost no problems.

Some points to be edited, such as using both "tumor" and "tumour".

Author Response

[ANSWER 1]:  The idea to compare the results with data on metastatic brain tumors would indeed be very interesting and a substantial added value for this study. Unfortunately, such data are not standard available in the database of the Belgian Cancer Registry and this is the same for most cancer registries worldwide (Riihimäki M, Thomsen H, Sundquist K, Sundquist J, Hemminki K. Clinical landscape of cancer metastases. Cancer Med. 2018 Nov;7(11):5534-5542. doi: 10.1002/cam4.1697. Epub 2018 Oct 16. PMID: 30328287; PMCID: PMC6246954.)

[ANSWER 2 ]:  Thank you for noticing these errors regarding the use of US and UK English. As suggested we corrected the text accordingly for consistency (to US English):

  • Line 42 on page 1: “For all malignant brain tumours together,…” corrected to “For all malignant brain tumors together,…”
  • Line 50 on page 2: “Keywords: COVID-19, brain tumours, …” corrected to “Keywords: COVID-19, brain tumors, …”
  • Line 192 on page 8: “Single centre studies…” corrected to “Single center studies…”
  • Line 245 on page 9: “…influenced survival in brain tumour patients, …” corrected to “…influenced survival in brain tumor patients, …”

Reviewer 3 Report

Comments and Suggestions for Authors

Analysis of COVID-19 epidemic related close-down of standard medical care system is presented. Results are presented in a statistically correct manner, but can be intuitively prognosed: close-down of a medical system is having a negative impact to survival of highly malignant tumors. On the other hand, this must be documented. An interesting factor, that was not addressed in this analysis - what was the impact of directly COVID-19 related deaths in this group. Maybe it can be extrapolated from the National Cancer Registry? Data can add a very interesting aspect to the survival.

Author Response

[ANSWER]:  We would like to thank reviewer 3 for addressing this highly relevant additional factor.  

Data on the number of deaths specifically due to COVID-19 infections in our study population (source: death certificates) are currently too incomplete to be added to the manuscript. However, based on administrative information from reimbursement data we were able to quantify the proportion of patients in the study population with a confirmed COVID-19 infection during hospitalisation. These data only reflect the hospital setting and thus a subset of COVID-19 infections, but can give an idea of the minimal proportion of patients affected by COVID-19 during follow-up of our study. Thus, such results can be added as nuancing factor in the discussion.  However, it has to be emphasized that these data, cannot be used for strong causal conclusions between COVID-19 infections and survival of our cancer cohort.

We added the following sentences in the discussion to elaborate on this topic after lines 244-246 on page 9, in which we introduced the topic in the previous version of the text:

“However, a last factor that might have influenced survival in brain tumor patients, is an intercurrent COVID-19 infection causing a delay in or suspension of treatment, or even death [21, 32]. In an earlier study of the Belgian Cancer Registry, we concluded that a substantial excess mortality was observed in the prevalent Belgian cancer cohort (diagnosed between 2013 and 2018 and alive on January 1, 2020) during the first wave of COVID-19, which, however, was not considerably different from the excess mortality in the general population [33]. Based on administrative information from reimbursement data, we found that at least 9.9% of our study population for 2020 suffered from a concomitant COVID-19 infection confirmed in a hospital-setting during the follow-up of our study (data not shown). Such COVID-19 infections can have an impact on the treatment trajectories and observed survival of these patients, but we cannot draw strong causal conclusions since cause of death information obtained from death certificates was unavailable at the time of this study. Additionally, it could also be possible that some brain tumor diagnoses were not even registered due to the excess mortality caused by the COVID-19 pandemic in the general population. Based on a statistical approximation using the number of deaths of the general Belgian population in 2020, this impact was estimated to be negligible in our study population (<1 missing diagnosis) [34].”

References added:

  1. Silversmit G, Verdoodt F, Van Damme N, De Schutter H, & Van Eycken L. Excess mortality in a nationwide cohort of cancer patients during the Initial phase of the COVID-19 pandemic in Belgium. Cancer Epidemiology, Biomarkers & Prevention. 2021; 30(9): 1615-1619.
  2. Directorate-general Statistics Belgium. Available online: http://www.statbel.fgov.be/ (accessed on 12 December 2023).